# Representational Rényi Heterogeneity

**DOI:** 10.3390/e22040417

**Published:** 2020-04-07

**Authors:** Abraham Nunes, Martin Alda, Timothy Bardouille, Thomas Trappenberg

**Affiliations:** 1Department of Psychiatry, Dalhousie University, Halifax, NS B3H 2E2, Canada; malda@dal.ca; 2Faculty of Computer Science, Dalhousie University, Halifax, NS B3H 4R2, Canada; 3Department of Physics and Atmospheric Sciences, Dalhousie University, Halifax, NS B3H 4R2, Canada; tim.bardouille@dal.ca

**Keywords:** heterogeneity, diversity, Rényi heterogeneity, representation learning, variational autoencoder, functional diversity indices, Hill numbers, Leinster–Cobbold Index, Rao’s quadratic entropy

## Abstract

A discrete system’s heterogeneity is measured by the Rényi heterogeneity family of indices (also known as Hill numbers or Hannah–Kay indices), whose units are the numbers equivalent. Unfortunately, numbers equivalent heterogeneity measures for non-categorical data require a priori (A) categorical partitioning and (B) pairwise distance measurement on the observable data space, thereby precluding application to problems with ill-defined categories or where semantically relevant features must be learned as abstractions from some data. We thus introduce representational Rényi heterogeneity (RRH), which transforms an observable domain onto a latent space upon which the Rényi heterogeneity is both tractable and semantically relevant. This method requires neither a priori binning nor definition of a distance function on the observable space. We show that RRH can generalize existing biodiversity and economic equality indices. Compared with existing indices on a beta-mixture distribution, we show that RRH responds more appropriately to changes in mixture component separation and weighting. Finally, we demonstrate the measurement of RRH in a set of natural images, with respect to abstract representations learned by a deep neural network. The RRH approach will further enable heterogeneity measurement in disciplines whose data do not easily conform to the assumptions of existing indices.

## 1. Introduction

Measuring heterogeneity is of broad scientific importance, such as in studies of biodiversity (ecology and microbiology) [1,2], resource concentration (economics) [3], and consistency of clinical trial results (biostatistics) [4], to name a few. In most of these cases, one measures the heterogeneity of a discrete system equipped with a probability mass function.

Discrete systems assume that all observations of a given state are identical (zero distance), and that all pairwise distances between states are permutation invariant. This assumption is violated when relative distances between states are important. For example, an ecosystem is not biodiverse if all species serve the same functional role [5]. Although species are categorical labels, their pairwise differences in terms of ecological functions differ and thus violate the discrete space assumptions. Mathematical ecologists have thus developed heterogeneity measures for non-categorical systems, which they generally call “functional diversity indices” [6,7,8,9,10,11]. These indices typically require a priori discretization and specification of a distance function on the observable space.

The requirement for defining the state space a priori is problematic when the states are incompletely observable: that is, when they may be noisy, unreliable, or invalid. For example, consider sampling a patient from a population of individuals with psychiatric disorders and assigning a categorical state label corresponding to his or her diagnosis according to standard definitions [12]. Given that psychiatric conditions are not defined by objective biomarkers, the individual’s diagnostic state will be uncertain. Indeed, many of these conditions are inconsistently diagnosed across raters [13], and there is no guarantee that they correspond to valid biological processes. Alternatively, it is possible that variation within some categorical diagnostic groups is simply related to diagnostic “noise,” or nuisance variation, but that variation within other diagnostic groups constitutes the presence of sub-strata. Appropriate measurement of heterogeneity in such disciplines requires freedom from the discretization requirement of existing non-categorical heterogeneity indices.

Pre-specified distance functions may fail to capture semantically relevant geometry in the raw feature space. For example, the Euclidean distance between Edmonton and Johannesburg is relatively useless since the straight-line path cannot be traversed. Rather, the appropriate distances between points must account for the data’s underlying manifold of support. Representation learning addresses this problem by learning a latent embedding upon which distances are of greater semantic relevance [14]. Indeed, we have observed superior clustering of natural images embedded on Riemannian manifolds [15] (but also see Shao et al. [16]), and preservation of semantic hierarchies when linguistic data are embedded on a hyperbolic space [17].

Therefore, we seek non-categorical heterogeneity indices without requisite a priori definition of categorical state labels or a distance function. The present study proposes a solution to these problems based on the measurement of heterogeneity on learned latent representations, rather than on raw observable data. Our method, representational Rényi heterogeneity (RRH), involves learning a mapping from the space of observable data to a latent space upon which an existing measure (the Rényi heterogeneity [18], also known as the Hill numbers [19] or Hannah–Kay indices [20]) is meaningful and tractable.

The paper is structured as follows. Section 2 introduces the original categorical formulation of Rényi heterogeneity and various approaches by which it has been generalized for application on non-categorical spaces [8,10,21]. Limitations of these indices are highlighted, thereby motivating Section 3, which introduces the theory of Representational Rényi Heterogeneity (RRH), which generalizes the process for computing many indices of biodiversity and economic equality. Section 4 provides an illustration of how RRH may be measured in various analytical contexts. We provide an exact comparison of RRH to existing non-categorical heterogeneity indices under a tractable mixture of beta distributions. To highlight the generalizability of our approach to complex latent variable models, we also provide an evaluation of RRH applied to the latent representations of a handwritten image dataset [22] learned by a variational autoencoder [23,24]. Finally, in Section 5 we provide a summary of our findings and discuss avenues for future work.

## 2. Existing Heterogeneity Indices

### 2.1. Rényi Heterogeneity in Categorical Systems

There are many approaches to derive Rényi heterogeneity [18,19,20]. Here, we loosely follow the presentation of Eliazar and Sokolov [25] by using the metaphor of repeated sampling from a discrete system *X* with event space $X=\left\{1,2,\ldots ,n\right\}$ and probability distribution $p={\left(p_{i}\right)}_{i=1,2,\ldots ,n}$. The probability that $q\in N_{>1}$ independent and identically distributed (i.i.d.) realizations of *X*, sampled with replacement, will be identical is
(1)$$P_{X}\left[x_{1}=x_{2}=\cdots =x_{q}\right]=\sum_{i=1}^{n}p_{i}^{q}.$$

Now let $X_{*}$ be an idealized reference system with a uniform probability distribution over $n_{*}$ categorical states, $p_{*}={\left(n_{*}^{-1}\right)}_{i=1,2,\ldots ,n_{*}}$, and let $\left(x_{*1},x_{*2},\ldots ,x_{*q}\right)$ be a sample of *q* i.i.d. realizations of $X_{*}$ such that
(2)$$P_{X}\left[x_{1}=x_{2}=\cdots =x_{q}\right]=P_{X_{*}}\left[x_{*1}=x_{*2}=\cdots =x_{*q}\right]=\sum_{i=1}^{n_{*}}n_{*}^{-q}.$$

We call $X_{*}$ an “idealized” categorical system because its probability distribution is uniform, and it is a “reference” system for *X* in that the probability of drawing homogeneous samples of *q* observations from both systems is identical. Substituting Equation (Equation 2) into Equation (Equation 1) and solving for $n_{*}$ yields the Rényi heterogeneity of order *q*,
(3)$$\Pi_{q}\left(p\right)={\left(\sum_{i=1}^{n}p_{i}^{q}\right)}^{\frac{1}{1-q}}=n_{*},$$
whose units are the numbers equivalent of system *X* [1,26,27,28], insofar as $n_{*}$ is the number of states in an “equivalent” (idealized reference) system $X_{*}$. Thus far, we have restricted the parameter *q* to take integer values greater than 1 solely to facilitate this intuitive derivation in a concise fashion. However, the elasticity parameter *q* in Equation (Equation 3) can be any real number (but q≠1), although in the context of heterogeneity measurement only q≥0 are used [1,25]. Despite Equation (Equation 3) being udefined at q=1 directly, L’Hôpital’s rule can be used to show that the limit q→1 exists, wherein it corresponds to the exponential of Shannon’s entropy [28,29], known as perplexity [30].

Equation (Equation 3) is the exponential of Rényi’s entropy [18], and is alternatively known as the Hill numbers in ecology [1,19], Hannah–Kay indices in economics [20], and generalized inverse participation ratio in physics [25]. Interestingly, it generalizes or can be transformed into several heterogeneity indices that are commonly employed across scientific disciplines (Table 1).

#### 2.1.1. Properties of the Rényi Heterogeneity

Equation (Equation 3) satisfies several properties that render it a preferable measure of heterogeneity. These have been detailed elsewhere [1,20,25,28,33,38], but we focus on three properties that are of particular relevance for the remainder of this paper.

First, $\Pi_{q}$ satisfies the principle of transfers [39,40] which states that any equality-increasing transfer of probability between states must increase the heterogeneity. The maximal value of $\Pi_{q}$ is attained if and only if $p_{i}=p_{j}$ for all (i,j)∈{1,2,…,n}. This property follows from Schur-concavity of Equation (Equation 3) [20].

Second, $\Pi_{q}$ satisfies the replication principle [1,38,41], which is equivalent to stating that Equation (Equation 3) scales linearly with the number of equally probable states in an idealized categorical system [25]. More formally, consider a set of systems $X_{1},X_{2},\ldots ,X_{N}$ with probability distributions $p_{1},p_{2},\ldots ,p_{N}$ over respective discrete event spaces $X_{1},X_{2},\ldots ,X_{N}$. These systems are also assumed to satisfy the following properties:Event spaces are disjoint: $X_{i}\cap X_{j}=\emptyset$ for all (i,j)∈{1,2,…,N} where i≠jAll systems have equal heterogeneity: $\Pi_{q}\left(p_{1}\right)=\Pi_{q}\left(p_{2}\right)=\cdots =\Pi_{q}\left(p_{i}\right)=\cdots =\Pi_{q}\left(p_{N}\right)$

The replication principle states that if we combine $X_{1},X_{2},\ldots ,X_{N}$ into a pooled system *X* with probability distribution $\bar{p}$, then
(4)$$\Pi_{q}\left(\bar{p}\right)=N\Pi_{q}\left(p_{i}\right)$$
must hold (see Appendix B for proof that Rényi heterogeneity satisfies the replication principle).

The replication principle suggests that Equation (Equation 3) satisfies a property known as decomposability, in that the heterogeneity of a pooled system can be decomposed into that arising from variation within and between component subsystems. However, we require that this property be satisfied when either (A) subsystems’ event spaces are overlapping, or (B) subsystems do not have equal heterogeneity. The decomposability property will be particularly important for Section 3, and so we detail it further in Section 2.1.2.

#### 2.1.2. Decomposition of Categorical Rényi Heterogeneity

Consider a system *X* defined by pooling subsystems $X_{1},X_{2},\ldots ,X_{N}$ with potentially overlapping event spaces $X_{1},X_{2},\ldots ,X_{N}$, respectively. The event space of the pooled system is defined as
(5)$$X=\cup_{i=1}^{N}X_{i}=\left\{1,2,\ldots ,n\right\}.$$

Furthermore, we define the matrix $P={\left(p_{ij}\right)}_{i=1,2,\ldots ,N}^{j=1,2,\ldots ,n}$ whose *i*^th^ row is the probability of system $X_{i}$ being observed in each state j∈{1,2,…,n}.

It may be the case that some subsystems comprise a larger proportion of *X* than others. For instance, if the probability distribution for subsystem $X_{i}$ was estimated based on a larger sample size than that of $X_{j}$, one may want to weight the contribution of $X_{i}$ higher. Thus, we define a column vector of weights $w={\left(w_{i}\right)}_{i=1,2,\ldots ,N}$ over the *N* subsystems such that $\sum_{i=1}^{N}w_{i}=1$ and $w_{i}\geq 0$ for all *i*. The probability distribution over states in the pooled system *X* may thus be computed as $\bar{p}=\sum_{i=1}^{N}w_{i}p_{i}$, from which the definition of pooled heterogeneity follows:(6)$$\Pi_{q}^{P}\left(P,w\right)={\left[\sum_{j=1}^{n}{\left(\sum_{i=1}^{N}w_{i}p_{ij}\right)}^{q}\right]}^{\frac{1}{1-q}}.$$

One can interpret $\Pi_{q}^{P}\left(P,w\right)$ as the effective number of states in the pooled categorical system *X*.

Jost [28] showed that the within-group heterogeneity, which is the effective number of unique states arising from individual component systems, can be defined as
(7)$$\Pi_{q}^{W}\left(P,w\right)={\left[\frac{\sum_{i=1}^{N}w_{i}^{q}\left(\sum_{j=1}^{n}p_{ij}^{q}\right)}{\sum_{k=1}^{N}w_{k}^{q}}\right]}^{\frac{1}{1-q}},$$

For example, in the case where all subsystems have disjoint event spaces with heterogeneity equal to constant ν, then they each contribute ν unique states to the pooled system *X*.

Deriving the between-group heterogeneity $\Pi_{q}^{B}\left(P,w\right)$, is thus straightforward. If the effective total number of states in the pooled system is $\Pi_{q}^{P}\left(P,w\right)$, and the effective number of unique states contributed by distinct subsystems is $\Pi_{q}^{W}\left(P,w\right)$, then
(8)$$\Pi_{q}^{B}\left(P,w\right)=\frac{\Pi_{q}^{P}\left(P,w\right)}{\Pi_{q}^{W}\left(P,w\right)}$$
is the effective number of completely distinct subsystems in the pooled system *X*. A word of caution is warranted. If we require that within-group heterogeneity is a lower bound on pooled heterogeneity [42], then (Jost [28], see Proofs 2 and 3) showed that Equation (Equation 8) will hold (A) at any value of *q* when weights are equal (i.e., $w_{i}=1/N$ for all i∈{1,2,…,N}), or (B) only at q=0 and q=1 if weights are unequal.

#### 2.1.3. Limitations of Categorical Rényi Heterogeneity

The chief limitation of Rényi heterogeneity (Equation (Equation 3)) is its assumption that all states in a system *X* (with event space X={1,2,…,n} and probability distribution $p={\left(p_{i}\right)}_{i=1,2,\ldots ,n}$) are categorical. More formally, the dissimilarity between a pair of observations (x,y)∈X from this system is defined by the discrete metric
(9)$$d^{*}\left(x,y\right)=1-\delta_{xy},$$
where $\delta_{xy}$ is Kronecker’s delta, which takes a value of 1 if x=y and 0 otherwise. Since the discrete metric assumption is an idealization, we have continued to use the asterisk to qualify an arbitrary distance function d(·,·) as categorical in nature. The resulting expected pairwise distance matrix between states in *X* is
(10)$$D^{*}={\left[d^{*}\left(i,j\right)\right]}_{i=1,2,\ldots ,n}^{j=1,2,\ldots ,n}=1\;1^{⊤}-I,$$
where $1={\left(1\right)}_{i=1,2,\ldots ,n}$ is a column vector of ones, and $I={\left(\delta_{ij}\right)}_{i=1,2,\ldots ,n}^{j=1,2,\ldots ,n}$ is the n×n identity matrix.

Clearly, many systems of interest in the real world are not categorical. For example, although we may label a sample of organisms according to their respective species, there may be differences between these taxonomic classes that are relevant to the functioning of the ecosystem as a whole [5]. It is also possible that no valid and reliable set of categorical labels is known a priori for a system whose event space is naturally non-categorical.

### 2.2. Non-Categorical Heterogeneity Indices

Consider a system *X* with probability distribution $p={\left(p_{i}\right)}_{i=1,2,\ldots ,n}$ defined over event space X={1,2,…,n} and equipped with dissimilarity function $d_{X}\left(\cdot ,\cdot \right)$. We assume that $d_{X}$ is more general than the discrete metric (Equation (Equation 9)), and further still need not be a true (metric) distance. For such systems, there are three heterogeneity indices whose units are numbers equivalent, and respect the replication principle [6,8,10,11,21]. Much like our derivation of the Rényi heterogeneity in Section 2.1, these indices quantify the heterogeneity of a non-categorical system as the number of states in an idealized reference system, but differ primarily in how the idealized reference is defined. We begin with a discussion of the Numbers-Equivalent Quadratic Entropy (Section 2.2.1), followed by the Functional Hill Numbers (Section 2.2.2) and the Leinster–Cobbold index [10] (Section 2.2.3).

#### 2.2.1. Numbers Equivalent Quadratic Entropy

Rao [43] introduced the diversity index commonly known as Rao’s quadratic entropy (RQE),
(11)$$Q_{1}\left(D,p\right)=\sum_{i=1}^{n}\sum_{j=1}^{n}D_{ij}p_{i}p_{j}$$
where D is an n×n matrix where $D_{ij}=d_{X}\left(i,j\right)$ for states (i,j)∈X.

Ricotta and Szeidl [21] assume that $D_{ij}=1$ means that states *i* and *j* are maximally dissimilar (i.e., categorically different), and that $D_{ij}=0$ means i=j, which occurs when X is a categorical system. An arbitrary dissimilarity matrix D can be rescaled to respect this assumption by applying the following transformation:(12)$$\tilde{D}=\frac{D-\min_{ij}D_{ij}}{\max_{ij}D_{ij}-\min_{ij}D_{ij}}.$$

Under this transformation, Ricotta and Szeidl [21] search for an idealized categorical reference system $X_{*}$ with event space $X_{*}=\left\{1,2,\ldots ,n_{*}\right\}$, probability distribution $p_{*}={\left(n_{*}^{-1}\right)}_{i=1,2,\ldots ,n_{*}}$, and RQE equal to that of *X*. For a column vector of ones, $1={\left(1\right)}_{i=1,2,\ldots ,n_{*}}$, and the identity matrix $I={\left(\delta_{ij}\right)}_{i=1,2,\ldots ,n_{*}}^{j=1,2,\ldots ,n_{*}}$, this is
(13)$$Q_{1}\left(\tilde{D},p\right)=Q_{1}\left(11^{⊤}-I,p_{*}\right).$$

Expanding the right-hand side, we have
(14)$$Q_{1}\left(\tilde{D},p\right)=\sum_{i=1}^{n_{*}}\sum_{j=1}^{n_{*}}n_{*}^{-2}\left(1-\delta_{ij}\right)=1-\frac{1}{n_{*}}.$$

Recalling that $\Pi_{q}\left(p_{*}\right)=n_{*}$ and substituting into Equation (Equation 14) yields
(15)$$\Pi_{q}\left(p_{*}\right)={\left[1-Q_{1}\left(\tilde{D},p\right)\right]}^{-1},$$
which establishes the units of ${\left[1-Q_{1}\left(\tilde{D},p\right)\right]}^{-1}$ as numbers equivalent.

For consistency, we require that $\Pi_{q}\left(p_{*}\right)=\Pi_{q}\left(p\right)$ if $\tilde{D}$ were categorical. This only holds at q=2:(16)$${\left[1-Q_{1}\left(\tilde{D},p\right)\right]}^{-1}={\left[1-\sum_{i=1}^{n}\sum_{j=1}^{n}p_{i}p_{j}\left(1-\delta_{ij}\right)\right]}^{-1}={\left(\sum_{i=1}^{n}p_{i}^{2}\right)}^{-1}=\Pi_{2}\left(p_{*}\right).$$

Based on this result, Ricotta and Szeidl [21] define the numbers equivalent quadratic entropy ${\hat{Q}}_{e}$ as
(17)$${\hat{Q}}_{e}\left(\tilde{D},p\right)={\left(1-Q_{1}\left(\tilde{D},p\right)\right)}^{-1}.$$

This can be interpreted as the inverse Simpson concentration of an idealized categorical reference system whose average pairwise distance between states is equal to $Q_{1}\left(\tilde{D},p\right)$.

#### 2.2.2. Functional Hill Numbers

Chiu and Chao [8] derived the Functional Hill Numbers, denoted $F_{q}$, based on a similar procedure to that of Ricotta and Szeidl [21]. However, whereas ${\hat{Q}}_{e}$ uses a purely categorical system as the idealized reference, $F_{q}$ requires only that
(18)$$Q_{1}\left(D,p\right)=\sum_{i=1}^{n_{*}}\sum_{j=1}^{n_{*}}Q_{1}\left(D,p\right)p_{*i}p_{*j}=\sum_{i=1}^{n_{*}}\sum_{j=1}^{n_{*}}Q_{1}\left(D,p\right)n_{*}^{-2},$$
which means that the idealized reference system is one for which the between-state distance matrix is set to $Q_{1}\left(D,p\right)$ everywhere (or to 0 along the leading diagonal and $Q_{1}\left(D,p\right)n_{*}/\left(n_{*}-1\right)$ on the off diagonals).

Chiu and Chao [8] generalized Rao’s quadratic entropy to include the elasticity parameter q≥0
(19)$$Q_{q}\left(D,p\right)=\sum_{i=1}^{n}\sum_{j=1}^{n}D_{ij}{\left(p_{i}p_{j}\right)}^{q},$$
and sought to find $n_{*}$ for the idealized reference system satisfying Equation (Equation 18) and the following:(20)$$Q_{q}\left(D,p\right)=\sum_{i=1}^{n_{*}}\sum_{j=1}^{n_{*}}Q_{1}\left(D,p\right){\left(\frac{1}{n_{*}}\frac{1}{n_{*}}\right)}^{q}.$$

Solving Equation (Equation 20) for $n_{*}$ yields the functional Hill numbers of order *q*:(21)$$F_{q}\left(D,p\right)={\left(\frac{Q_{q}\left(D,p\right)}{Q_{1}\left(D,p\right)}\right)}^{\frac{1}{2(1-q)}}=n_{*},$$
which is the effective number of states in an idealized categorical reference system whose distance function is scaled by a factor of $Q_{1}\left(D,p\right)n_{*}/\left(n_{*}-1\right)$.

#### 2.2.3. Leinster–Cobbold Index

The index derived by Leinster and Cobbold [10], denoted $L_{q}$, is distinct from ${\hat{Q}}_{e}$ and $F_{q}$ in two ways. First, for a given system *X*, the $L_{q}$ is not derived based on finding an idealized reference system $X_{*}$ whose average between-state dissimilarity is equal to that of *X*. Second, it does not use a dissimilarity matrix; rather, it uses a measure of similarity or affinity.

The Leinster–Cobbold index may be derived by simple extension of Equation (Equation 3). Assuming *X* has state space X={1,2,…,n} with probability distribution $p={\left(p_{i}\right)}_{i=1,2,\ldots ,n}$, we note that
(22)$$\Pi_{q}\left(p\right)={\left(\sum_{i=1}^{n}p_{i}^{q}\right)}^{\frac{1}{1-q}}={\left[\sum_{i=1}^{n}p_{i}{\left(Ip\right)}_{i}^{q-1}\right]}^{\frac{1}{1-q}}.$$

Here, I is the n×n identity matrix representing the pairwise similarities between states in *X*. The Leinster–Cobbold index generalizes I to be any n×n similarity matrix S, yielding the following formula:(23)$$L_{q}\left(S,p\right)={\left[\sum_{i=1}^{n}p_{i}{\left(\sum_{j=1}^{n}S_{ij}p_{j}\right)}^{q-1}\right]}^{\frac{1}{1-q}}.$$

The similarity matrix can be obtained from a dissimilarity matrix by the transformation $S_{ij}=e^{-uD_{ij}}$, where u≥0 is a scaling factor. When u=0, then S is 1 everywhere. Conversely, when u→∞, then S approaches I. The Leinster–Cobbold index can thus be interpreted as an effective number if the states are in an idealized reference system (i.e., one with uniform probabilities over states) whose topology is also governed by the similarity matrix S.

#### 2.2.4. Limitations of Existing Non-Categorical Heterogeneity Indices

We illustrate several limitations of the ${\hat{Q}}_{e}$, $F_{q}$, and $L_{q}$ indices using a simple 3-state system *X* with event space X={1,2,3} over which we specify a probability distribution
(24)$$p\left(\kappa \right)=\left\{\begin{array}{cc} {\left(1,0,0\right)}^{⊤} & \kappa =0\\ {\left(\frac{1}{3},\frac{1}{3},\frac{1}{3}\right)}^{⊤} & \kappa =1\\ {\left(0,0,1\right)}^{⊤} & \kappa =\infty \\ {\left(\frac{1}{1+\sqrt{\kappa }+\kappa },\frac{\sqrt{\kappa }}{1+\sqrt{\kappa }+\kappa },\frac{\kappa }{1+\sqrt{\kappa }+\kappa }\right)}^{⊤} & \mathrm{Otherwise} \end{array}\right.$$
where 0≤κ is a parameter that smoothly varies the level of inequality. When κ=1 the distribution is perfectly even (Figure 1A). Since an undirected graph of the system is arranged in a triangle with height *h* and base *b*, we also specify the following parametric distance matrix,
(25)$$D\left(h,b\right)=\left(\begin{array}{ccc} 0 & b & \sqrt{\frac{b^{2}}{4}+h^{2}}\\ b & 0 & \sqrt{\frac{b^{2}}{4}+h^{2}}\\ \sqrt{\frac{b^{2}}{4}+h^{2}} & \sqrt{\frac{b^{2}}{4}+h^{2}} & 0 \end{array}\right),$$
which allows us to smoothly vary the level of dissimilarity between states in *X*. Importantly, Equation (Equation 25) allows us to generate distance matrices that are either metric (when $h<b\sqrt{3}/2$; Definition 1) or ultrametric (when $h\geq b\sqrt{3}/2$; Definition 2). This is illustrated in Figure 1B.

**Definition** **1** (Metric distance).
*A function $d:X\times X\to R_{\geq 0}$ on a set X is a metric if and only if all of the following conditions are satisfied for all (x,y,z)∈X:*
*1* 
*Non-negativity: d(x,y)≥0*
*2* 
*Identity of indiscernibles: d(x,y)=0⇔x=y*
*3* 
*Symmetry: d(x,y)=d(y,x)*
*4* 
*Triangle inequality: d(x,z)≤d(x,y)+d(y,z)*



**Definition** **2** (Ultrametric distance).
*A function $d:X\times X\to R_{\geq 0}$ on a set X is ultrametric if and only if, for all (x,y,z)∈X, criteria 1-3 for a metric are satisfied (Definition 1), in addition to the ultrametric triangle inequality:*
(26)$$d\left(x,z\right)\leq \max\left\{d(x,y),d(y,z)\right\}$$


Figure 1C compares the ${\hat{Q}}_{e},F_{q}$, and $L_{q}$ indices when applied to *X* across variation in between-state distances (via Equation (Equation 25)) and skewness in the probability distribution over states (Equation (Equation 24)). With respect to the numbers equivalent quadratic entropy (${\hat{Q}}_{e}$; Section 2.2.1), we note that its behavior is categorically different with respect to whether the distance matrix is ultrametric. That is ${\hat{Q}}_{e}$ increases with the triangle height parameter *h* (Equation (Equation 25)) until it passes the ultrametric threshold, after which it decreases monotonically with *h*. The behavior of ${\hat{Q}}_{e}$ is sensible in the ultrametric range. When the distance matrix is scaled, as in Equation (Equation 12), pulling one of the three states in *X* further away from the remaining two should function similarly to progressively merging the latter states. Thus, the behavior of ${\hat{Q}}_{e}$ is highly sensitive to whether a given distance matrix is ultrametric (which will often not be the case in real-world applications).

With respect to $F_{q}$, a notable benefit in comparison to ${\hat{Q}}_{e}$ is that $F_{q}$ behaves consistently regardless of whether distance is ultrametric. However, Figure 1 shows other drawbacks. First, we can see that $F_{q}$ becomes insensitive to D(h,1) when p(κ) is perfectly even (shown analytically in Appendix B). Second, $F_{q}$ can paradoxically estimate a greater number of states than the theoretical maximum allows. That this occurs when the state probability distribution is more unequal violates the principle of transfers [20,33,39,40] (Section 2.1.1). This is made more problematic since Figure 1C shows it occurs when one state is being pushed closer to the others (i.e., with smaller values of *h*). To summarize, the functional Hill numbers are estimating more states than are really present despite the reduction in between-state distances and greater inequality in the probability mass function.

Figure 1C shows that the Leinster-Cobbold index compares favorably to $F_{q}$ because the former does not lose sensitivity to dissimilarity when p(κ) is perfectly even. However, Figure 1D shows that the Leinster-Cobbold index is particularly sensitive to the form of similarity transformation. In the present case, the maximal value of the $L_{q}$ gradually approaches 3 as *u* grows (and only when u→∞ does it reach 3), while progressively losing sensitivity to distance. As mentioned by Leinster and Cobbold [10], the choice of *u* or other similarity transformation is dependent on the importance assigned to functional differences between states. However, it is not clear how a given similarity transformation (e.g., *u*), and therefore the idealized reference system of $L_{q}$, should be validated.

Above all of the idiosyncratic limitations of existing numbers equivalent heterogeneity indices, we must highlight two basic assumptions they all share. First, they continue to assume that some valid and reliable categorical partitioning on *X* is known a priori. Second, they assume that a distance function specified a priori describes semantically relevant geometry of the system in question. These two limitations are not independent, since an unreliable categorical partitioning of the state space will lead to erroneous estimates of the pairwise distances between states. Thus, we seek an approach for measuring heterogeneity that has neither these limitations, nor those shown above to be specific to the other numbers equivalent heterogeneity indices for non-categorical systems.

## 3. Representational Rényi Heterogeneity

In this section, we propose an alternative approach to the indices of Section 2.2 that we call representational Rényi heterogeneity (RRH). It involves transforming *X* into a representation *Z*, defined on an unobservable or latent event space Z, that satisfies two criteria:The representation *Z* captures the semantically relevant variation in *X*Rényi heterogeneity can be directly computed on *Z*

Satisfaction of the first criterion can only be ascertained in a domain-specific fashion. Since *Z* is essentially a model of *X*, investigators must justify that this model is appropriate for the scientific question at hand. For example, an investigator may evaluate the ability of *X* to be reconstructed from representation *Z* under cross-validation. The second criterion simply means that the transformation of X→Z must specify a probability distribution on Z upon which the Rényi heterogeneity can be directly computed.

Figure 2 illustrates the basic idea of RRH. However, the specifics of this framework differ based on the topology of the representation *Z*. Thus, the remainder of this section discusses the following approaches:A.Application of standard Rényi heterogeneity (Section 2.1) when *Z* is a categorical representationB.Deriving parametric forms for Rényi heterogeneity when *Z* is a non-categorical representation

### 3.1. Rényi Heterogeneity on Categorical Representations

Let *X* be a system defined on an observable space X that is non-categorical and $n_{x}$-dimensional. Consider the scenario in which the semantically relevant variation in *X* is categorical: for instance, images of different object categories stored in raw form as real-valued vectors. An investigator may be interested in measuring the effective number of states in *X* with respect to this categorical variation. This requires transforming *X* into a semantically relevant categorical representation *Z* upon which Equation (Equation 3) can be applied.

Assume we have a large random sample of *N* points $X={\left(x_{i}\right)}_{i=1,2,\ldots ,N}$ from system *X*. We can conceptualize each discrete observation $x_{i}$ in this sample as the single point in the event space of a perfectly homogeneous subsystem $X_{i}$. When pooled, the subsystems ${\left\{X_{i}\right\}}_{i=1,2,\ldots ,N}$ constitute *X*. The contribution weights of each subsystem to *X* as a whole are denoted $w={\left(w_{i}\right)}_{i=1,2,\ldots ,N}$, where $\sum_{i=1}^{N}w_{i}=1$ and $w_{i}\geq 0$.

We now specify a vector-valued function f:X→P(Z) such that $x\mapsto f\left(x\right)={\left[f_{j}\left(x\right)\right]}_{j=1,2,\ldots ,n_{z}}$ is a mapping from $n_{x}$-dimensional coordinates on the observable space, x∈X, onto an $n_{z}$-dimensional discrete probability distribution over $Z=\{1,2,\ldots ,n_{z}\}$. Thus, $f(x_{i})$ can be conceptualized as mapping subsystem $X_{i}$ onto its categorical representation $Z_{i}$. After defining f, the effective number of states in the latent representation of $X_{i}$ can be computed as
(27)$$\Pi_{q}\left(x_{i}\right)={\left(\sum_{j=1}^{n_{z}}f_{j}^{q}\left(x_{i}\right)\right)}^{\frac{1}{1-q}}.$$

When $\Pi_{q}\left(x_{i}\right)=1$, then f assigns x to a single category with perfect certainty. Conversely, when $\Pi_{q}\left(x_{i}\right)=n_{z}$, then either $x_{i}$ belongs to all categorical states with equal probability, or f is maximally uncertain about the mapping of point $x_{i}$.

Mapping all points X onto the categorical latent space yields a collection of subsystems ${\left\{Z_{i}\right\}}_{i=1,2,\ldots ,N}$, which generate *Z* when pooled. Using Equation (Equation 6), we can compute the effective number of total states in *Z* as the pooled heterogeneity:(28)$$\Pi_{q}^{P}\left(X,w\right)={\left[\sum_{j=1}^{n_{z}}{\left(\sum_{i=1}^{N}w_{i}f_{j}\left(x_{i}\right)\right)}^{q}\right]}^{\frac{1}{1-q}},$$

Unfortunately, $\Pi_{q}^{P}\left(X,w\right)$ counts some heterogeneity that is due to uncertainty in the model (i.e., that quantified by Equation (Equation 27)). We, therefore, compute the effective number of states in *Z* per point x∈X using the within-group heterogeneity formula (Equation (Equation 7)):(29)$$\Pi_{q}^{W}\left(X,w\right)={\left[\frac{\sum_{i=1}^{N}w_{i}^{q}\left(\sum_{j=1}^{n_{z}}f_{j}^{q}\left(x_{i}\right)\right)}{\sum_{k=1}^{N}w_{k}^{q}}\right]}^{\frac{1}{1-q}}.$$

Finally, the effective number of states (points) in *X*—with respect to the categorical variation modeled by *Z*—can then be computed using the between-group heterogeneity formula (Equation (Equation 8)):(30)$$\Pi_{q}^{B}\left(X,w\right)=\frac{\Pi_{q}^{P}\left(X,w\right)}{\Pi_{q}^{W}\left(X,w\right)}.$$

Example 1 demonstrates that current methods of measuring biodiversity and wealth concentration can be viewed as special cases of categorical RRH.

**Example** **1** (Classical measurement of biodiversity and economic equality as categorical RRH).
*Definitions necessary for this example are shown in Table 2. The traditional analysis of species diversity and economic equality can be recovered from an RRH-based formulation when f is assumed to be deterministic and $w={\left(N^{-1}\right)}_{i=1,2,\ldots ,N}$. In this case within-group heterogeneity can be shown to reduce to 1:*
(31)$$\begin{array}{cc} \Pi_{q}^{W}\left(X,w\right)\; & ={\left[\sum_{i=1}^{N}\frac{N^{-q}}{\sum_{k=1}^{N}N^{-q}}\left(\sum_{j=1}^{n_{z}}f_{j}^{q}\left(x_{i}\right)\right)\right]}^{\frac{1}{1-q}}\\ \; & ={\left[\sum_{i=1}^{N}N^{-1}\left(1\right)\right]}^{\frac{1}{1-q}}\\ \; & =1. \end{array}$$

*Thus, we have*
(32)$$\begin{array}{cc} \Pi_{q}^{B}\left(X,w\right)\; & =\Pi_{q}^{P}\left(X,w\right)\\ \; & ={\left[\sum_{j=1}^{n_{z}}{\left(\sum_{i=1}^{N}N^{-1}f_{j}\left(x_{i}\right)\right)}^{q}\right]}^{\frac{1}{1-q}}\\ \; & ={\left[\sum_{j=1}^{n_{z}}{\left(\frac{N_{j}}{N}\right)}^{q}\right]}^{\frac{1}{1-q}}, \end{array}$$
*which yields the categorical Rényi heterogeneity (Hill numbers for biodiversity analysis and Hannah–Kay indices in the economic setting [19,20]), and by extension many diversity indices to which it is connected (Table 1). Thus, traditional analysis of species biodiversity and economic equality are special cases of representational Rényi heterogeneity where the representation is specified by a mapping onto degenerate distributions over categorical labels. The only differences lie in the definition of observable and latent spaces, and the representational models.*

*In the case of biodiversity analysis, the model f in real-world practice may simply be a human expert assigning species labels to a sample of organisms from a field study. In the economic setting, one may speculate that f would essentially reduce to contracts specifying ownership of assets, whose value is deemed by market forces.*


### 3.2. Rényi Heterogeneity on Non-Categorical Representations

In Section 3.1, we dealt with instances in which semantically relevant variation in *X* is categorical, such as when object categories are embedded in images stored as real-valued vectors. Here, we consider scenarios in which the semantically relevant information in an observable system *X* is non-categorical: for instance, where a piece of text contains information about semantic concepts best represented as real-valued “word vectors” [44,45]. Measuring the effective number of distinct states in *X* with respect to this continuous variation requires transforming *X* into a semantically relevant continuous representation *Z* upon which procedures analogous to those of Section 3.1 may be undertaken.

Let *Z* be defined on an $n_{z}$-dimensional event space $Z\subseteq R^{n_{z}}$ over which there exists a family of parametric probability distributions P(Z) of a form chosen by the experimenter. Let f:X→P(Z) be a model that performs the mapping x↦f(·|x) from a point x∈X on the observable space to a probability density on Z. For example, if P(Z) is the family of multivariate Gaussians, then $f\left(z|x_{i}\right)=N\left(z|\mu_{i},\Sigma_{i}\right)$, where $\mu_{i}$ and $\Sigma_{i}$ are the Gaussian mean and covariance functions at $x_{i}$, respectively. Given a sample $X={\left(x_{i}\right)}_{i=1,2,\ldots ,N}$, as in Section 3.1, we compute the continuous analogue of Equation (Equation 27) as follows
(33)$$\Pi_{q}\left(x_{i}\right)={\left(\int_{Z}f^{q}\left(z|x_{i}\right)\;dz\right)}^{\frac{1}{1-q}}.$$

This formula yields the effective size of the domain of a uniform distribution on $R^{n_{z}}$ whose Rényi heterogeneity is equal to $\Pi_{q}\left(x_{i}\right)$ (proof is given in Appendix B). Thus, it is possible for $\Pi_{q}\left(x_{i}\right)$ to be less than 1, though it will remain non-negative.

Similar to the procedure in Section 3.1, we now define a continuous version of the within-observation heterogeneity
(34)$$\Pi_{q}^{W}\left(X,w\right)={\left[\sum_{i=1}^{N}\frac{w_{i}^{q}}{\sum_{j=1}^{N}w_{j}^{q}}\int_{Z}f^{q}\left(z|x_{i}\right)\;dz\right]}^{\frac{1}{1-q}},$$
which estimates the effective size of the latent space occupied per observable point x∈X.

In order to compute the pooled heterogeneity $\Pi_{q}^{P}\left(X,w\right)$, the experimenter must specify the form of the pooled distribution, here denoted ${\bar{f}}_{w}$. The conceptually most simple approach is non-parametric, using a model average,
(35)$${\bar{f}}_{w}\left(z|X\right)=\sum_{i=1}^{N}w_{i}f\left(z|x_{i}\right),$$
whereby the pooled heterogeneity would be
(36)$$\Pi_{q}^{P}\left(X,w\right)={\left[\int_{Z}{\left(\sum_{i=1}^{N}w_{i}f\left(z|x_{i}\right)\right)}^{q}\;dz\right]}^{\frac{1}{1-q}}.$$

The integral in Equation (Equation 36) may often be analytically intractable and potentially difficult to solve accurately in high dimensions with numerical methods. Furthermore, some areas of Z may be assigned low probability by $f(z|x_{i})$ for all i∈{1,2,…,N}. This is not a problem as the sample X becomes infinitely large. However, with finite samples, it may be the case that some representational states in Z are unlikely simply because we have not sampled from the corresponding regions of X. An alternative to Equation (Equation 35) is therefore to specify a parametric pooled distribution
(37)$${\bar{f}}_{w}\left(\cdot |X\right)=\Xi_{f}\left(X,w\right),$$
where $\Xi_{f}$ is a deterministic function that combines $f(\cdot |x_{i})$ for i∈{1,2,…,N} into a valid probability density on Z. In this case, the pooled Rényi heterogeneity is simply
(38)$$\Pi_{q}^{P}\left(X,w\right)={\left(\int_{Z}{\bar{f}}_{w}^{q}\left(z|X\right)\;dz\right)}^{\frac{1}{1-q}}.$$

Using either Equation (Equation 36) or (Equation 38) as the pooled heterogeneity and Equation (Equation 34) as the within-group heterogeneity, the effective number of distinct states in *X*—with respect to the non-categorical representation *Z*—can then be computed using Equation (Equation 30).

Figure 3 demonstrates the difference between the parametric and non-parametric approaches to pooling for non-categorical RRH, and Example 2 demonstrates one approach to parametric pooling for a mixture of multivariate Gaussians.

**Example** **2** (Parametric pooling of multivariate Gaussian distributions).
*Let $X={\left(x_{i}\right)}_{i=1,2,\ldots ,N}$ be a sample of $n_{x}$-dimensional vectors from a system X with event space $X\subseteq R^{n_{x}}$. Let Z be a latent representation of X with $n_{z}$-dimensional event space $Z=R^{n_{z}}$. Let*
(39)$$f\left(z|x_{i}\right)=N\left(z|\mu_{i},\Sigma_{i}\right)$$
*be a model that returns a multivariate Gaussian density with mean $\mu_{i}$ and covariance $\Sigma_{i}$ given point $x_{i}\in X$. Finally, let $w={\left(w_{i}\right)}_{i=1,2,\ldots ,N}$ be weights assigned to each sample in X such that $w_{i}\geq 0$ and $\sum_{i=1}^{N}w_{i}=1$.*

*If one assumes that the pooled distribution over Z given the set of components $f\left(z|x_{1}\right),f\left(z|x_{2}\right),\ldots ,f\left(z|x_{N}\right)$ is itself a multivariate Gaussian,*
(40)$${\bar{f}}_{w}\left(z|X\right)=N\left(z|\mu_{*},\Sigma_{*}\right)$$
*with $n_{z}\times 1$ pooled mean,*
(41)$$\mu_{*}=\sum_{i=1}^{N}w_{i}\mu_{i}$$
*and $n_{z}\times n_{z}$ pooled covariance matrix*
(42)$$\Sigma_{*}=-\mu_{*}\mu_{*}^{⊤}+\sum_{i=1}^{N}w_{i}\left[\Sigma_{i}+\mu_{i}\mu_{i}^{⊤}\right],$$
*then the pooled heterogeneity $\Pi_{q}^{P}$ is therefore simply the Rényi heterogeneity of a multivariate Gaussian,*
(43)$$\Pi_{q}\left(\Sigma \right)=\left\{\begin{array}{cc} \mathrm{Undefined} & q=0\\ {\left(2\pi e\right)}^{\frac{n_{z}}{2}}\sqrt{\left|\Sigma \right|} & q=1\\ {\left(2\pi \right)}^{\frac{n_{z}}{2}}\sqrt{\left|\Sigma \right|} & q=\infty \\ {\left(2\pi \right)}^{\frac{n_{z}}{2}}q^{\frac{n_{z}}{2(q-1)}}\sqrt{\left|\Sigma \right|} & \mathrm{Otherwise} \end{array}\right.$$
*evaluated at $\Sigma_{*}$. The derivation is provided in Appendix B [46]. Equation (Equation 43) at $\Sigma_{*}$ is interpreted as the effective size of space Z occupied by the complete latent representation of X under model f.*

*The within-group heterogeneity can be obtained for the set of components ${\left[f(z|x_{i})\right]}_{i=1,2,\ldots ,N}$ by solving Equation (Equation 34) for the Gaussian densities, yielding:*
(44)$$\Pi_{q}^{W}\left(\Sigma_{1:N},w\right)=\left\{\begin{array}{cc} \mathrm{Undefined} & q=0\\ \exp\left\{\frac{1}{2}\left(n_{z}+\sum_{i=1}^{N}w_{i}\log\left|2\pi \Sigma_{i}\right|\right)\right\} & q=1\\ 0 & q=\infty \\ {\left(2\pi \right)}^{\frac{n_{z}}{2}}{\left(\sum_{i=1}^{N}\frac{{\bar{w}}_{i}^{q}{\left|\Sigma_{i}\right|}^{\frac{1}{2}}}{q^{\frac{n_{z}}{2}}}\right)}^{\frac{1}{1-q}} & \mathrm{Otherwise} \end{array}\right.,$$
*where we denote $\Sigma_{1:N}={\left\{\Sigma_{i}\right\}}_{i=1,2,\ldots ,N}$ for parsimony, and ${\bar{w}}_{i}=w_{i}{\left(\sum_{j=1}^{N}w_{j}^{q}\right)}^{-1/q}$. Equation (Equation 44) estimates the effective size of the $n_{z}$-dimensional representational space occupied per state x∈X.*

*The effective number of states in X with respect to the continuous representation Z is thus the between-group heterogeneity $\Pi_{q}^{B}$ which can be computed as the ratio $\Pi_{q}\left(\Sigma_{*}\right)/\Pi_{q}^{W}\left(\Sigma_{1:N},w\right)$. The properties of this decomposition—specifically the conditions under which $\Pi_{q}^{B}\geq 1$ (Lande’s requirement [28,42])—are discussed further elsewhere [46].*


## 4. Empirical Applications of Representational Rényi Heterogeneity

In this section, we demonstrate two applications of RRH under assumptions of categorical (Section 4.1) and continuous (Section 4.2) latent spaces. First, Section 4.1, uses a simple closed-form system consisting of a mixture of two beta distributions on the (0,1) interval to give exact comparisons of the behavior of RRH against that of existing non-categorical heterogeneity indices (Section 2.2). This experiment provides evidence that existing non-categorical heterogeneity indices can demonstrate counterintuitive behavior under various circumstances. Second, Section 4.2 demonstrates that RRH can yield heterogeneity measurements that are sensible and tractably computed, even for highly complex mappings f:X→P(Z). There, we use a deep neural network to compute the effective number of observations in a database of handwritten images with respect to compressed latent representations on a continuous space.

### 4.1. Comparison of Heterogeneity Indices Under a Mixture of Beta Distributions

Consider a system *X* with event space X on the open interval (0,1), containing an embedded, unobservable, categorical structure represented by the latent system *Z* with event space $Z=\left\{1,2\right\}$. The systems’ collective behavior is governed by the joint distribution of a beta mixture model (BMM),
(45)$$p\left(x,z\right)=𝟙\left[z=1\right]\left(1-\theta_{1}\right){\mathrm{Beta}}_{\theta_{2},\theta_{3}}\left(x\right)+𝟙\left[z=2\right]\theta_{1}{\mathrm{Beta}}_{\theta_{3},\theta_{2}}\left(x\right),$$
where ${\mathrm{Beta}}_{\alpha ,\beta }\left(x\right)$ is the probability density function for a beta distribution with shape parameters α,β, and $\theta =\left(\theta_{1},\theta_{2},\theta_{3}\right)$ are parameters. The indicator function 1[·] evaluates to 1 if its argument is true, and to 0 otherwise. The prior distribution is
(46)$$p\left(z\right)=𝟙\left[z=1\right]\left(1-\theta_{1}\right)+𝟙\left[z=2\right]\theta_{1},$$
and marginal probability of observable data is as follows (see Figure 4 for illustrations):(47)$$p\left(x\right)=\left(1-\theta_{1}\right){\mathrm{Beta}}_{\theta_{2},\theta_{3}}\left(x\right)+\theta_{1}{\mathrm{Beta}}_{\theta_{3},\theta_{2}}\left(x\right).$$

To facilitate exact comparisons between heterogeneity indices, below, let us assume we have a model f:X→P(Z) that maps an observation x∈X onto a degenerate distribution over Z:(48)$$f_{\theta }\left(z|x\right)=𝟙\left[z=1\right]𝟙\left[x\leq \tau \left(\theta \right)\right]+𝟙\left[z=2\right]𝟙\left[x>\tau \left(\theta \right)\right].$$

The subscripting of $f_{\theta }$ denotes that the model is optimized such that the threshold 0≤τ(θ)≤1 is the solution to
(49)$$p\left(z=1|x=\tau (\theta )\right)=p\left(z=2|x=\tau (\theta )\right),$$
which is
(50)$$\tau \left(\theta \right)=\left\{\begin{array}{cc} {\left[{\left(\theta_{1}^{-1}-1\right)}^{\frac{1}{2(\theta_{2}-\theta_{3})}}{\left(1-\theta_{1}\right)}^{\frac{1}{2(\theta_{2}-\theta_{3})}}\theta_{1}^{-\frac{1}{2(\theta_{2}-\theta_{3})}}+1\right]}^{-1} & \theta_{2}-\theta_{3}\neq 0\\ 0 & \left(\left(\theta_{2}=\theta_{3}\right)\wedge \left(\theta_{1}>\frac{1}{2}\right)\right)\\ 1 & \mathrm{Otherwise} \end{array}\right.$$

Under this model, the categorical RRH at point x∈X is
(51)$$\Pi_{q}\left(x\right)={\left(\sum_{i=1}^{2}f_{\theta }^{q}\left(z=i|x\right)\right)}^{\frac{1}{1-q}}={\left({𝟙}^{q}\left[x\leq \tau (\theta )\right]+{𝟙}^{q}\left[x>\tau (\theta )\right]\right)}^{\frac{1}{1-q}}=1.$$

The expected value of $f_{\theta }\left(z=2|x\right)$ with respect to the data generating distribution (Equation (Equation 47)) is
(52)$$\begin{array}{cc} {\bar{f}}_{\theta }\left(z=2\right)\; & =E_{x∼p(x)}\left[f_{\theta }\left(z=2|x\right)\right]\\ \; & =\int_{0}^{1}p\left(x\right)𝟙\left[x>\tau (\theta )\right]\;dx\\ \; & =\int_{\tau (\theta )}^{1}p\left(x\right)\;dx\\ \; & =\left(1-\theta_{1}\right)I_{x}^{1}\left(\theta_{2},\theta_{3}\right)+\theta_{1}I_{x}^{1}\left(\theta_{3},\theta_{2}\right), \end{array}$$
where $I_{x_{0}}^{x_{1}}\left(a,b\right)$ is the generalized regularized incomplete beta function (BetaRegularized[$x_{0},x_{1},a,b$] command in the Wolfram language and betainc($a,b,x_{0},x_{1}$,regularized=True) in Python’s mpmath package). Equation (Equation 52) implies that ${\bar{f}}_{\theta }\left(z=1\right)=1-{\bar{f}}_{\theta }\left(z=2\right)$. The pooled heterogeneity is thus expressed as a function of θ as follows:(53)$$\Pi_{q}^{P}\left(\theta \right)=\left\{\begin{array}{cc} \sum_{i=1}^{2}𝟙\left[{\bar{f}}_{\theta }\left(z=i\right)>0\right] & q=0\\ \exp\left\{-\sum_{i=1}^{2}{\bar{f}}_{\theta }\left(z=i\right)\log{\bar{f}}_{\theta }\left(z=i\right)\right\} & q=1\\ {\left(\max_{i}{\bar{f}}_{\theta }\left(z=i\right)\right)}^{-1} & q=\infty \\ {\left(\sum_{i=1}^{2}{\bar{f}}_{\theta }^{q}\left(z=i\right)\right)}^{\frac{1}{1-q}} & \mathrm{Otherwise} \end{array}\right..$$

As a function of θ, the within-group heterogeneity is
(54)$$\begin{array}{cc} \Pi_{q}^{W}\left(\theta \right) & ={\left[\int_{0}^{1}\frac{p^{q}\left(x\right)}{\int_{0}^{1}p^{q}\left(u\right)\;du}{\left(\sum_{i=1}^{2}f_{\theta }\left(z=i|x\right)\right)}^{q}\;dx\right]}^{\frac{1}{1-q}}\\ \; & ={\left[\int_{0}^{1}\frac{p^{q}\left(x\right)}{\int_{0}^{1}p^{q}\left(u\right)\;du}\left(1\right)\;dx\right]}^{\frac{1}{1-q}}\\ \; & =1, \end{array}$$
and therefore the between-group heterogeneity is $\Pi_{q}^{B}\left(\theta \right)=\Pi_{q}^{P}\left(\theta \right)$.

Analytic expressions for the existing non-categorical heterogeneity indices ${\hat{Q}}_{e}$ (Equation (Equation 17)), $F_{q}$ (Equation (Equation 21)), and $L_{q}$ (Equation (Equation 23)) were computed as “best-case” scenarios, as follows. First, the probability distributions over states for all expressions was the true prior distribution (Equation (Equation 46)). Distance matrices—and by extension, the similarity matrix for $L_{q}$—were computed using the closed-form expectation of the absolute distance between two beta-distributed random variables (see Appendix C and the Appendix A).

Figure 5 compares the categorical RRH against ${\hat{Q}}_{e}$, $F_{q}$, and $L_{q}$ for BMM distributions of varying degrees of separation, and across different mixture component weights ($0.5\leq \theta_{1}<1$). Without significant loss of generality, we show only those comparisons at q=1 (which excludes the numbers equivalent quadratic entropy), and q=2.

The most salient differences between these indices occur when the BMM mixture components completely overlap (i.e., at $\theta_{2}=\theta_{3}$). The RRH correctly identifies that there is effectively only one component, regardless of mixture weights. Only the Leinster–Cobbold index showed invariance to the mixture weights when $\theta_{2}=\theta_{3}$, but it could not correctly identify that data were effectively unimodal.

The other stark difference arose when the mixture components were furthest apart (here when $\theta_{2}=5$ and $\theta_{3}=20$). At this setting, the functional Hill numbers showed a paradoxical increase in the heterogeneity estimate as the prior distribution on components was skewed. The Leinster–Cobbold index was appropriately concave throughout the range of prior weights, but it never reached a value of 2 at its peak (as expected based on the predictions outlined in Section 2.2.3). Conversely, the RRH was always concave and reached a peak of 2 when both mixture components were equally probable.

### 4.2. Representational Rényi Heterogeneity is Scalable to Deep Learning Models

In this example, the observable system *X* is that of images of handwritten digits defined on an event space $X={\left[0,1\right]}^{784}$ of dimension $n_{x}=784$ (the black and white images are flattened from 28×28 pixel matrices into 784-dimensional vectors). Our sample $X={\left(x_{ij}\right)}_{i=1,2,\ldots ,N}^{j=1,2,\ldots ,784}$ from this space is the familiar MNIST training dataset [22] (Figure 6), which consists of N=60,000 images roughly evenly distributed across digits {0,1,…,9}, and where approximately 10% of all images come from each class. We assume each image carries equal importance, given by a weight vector $w={\left(N^{-1}\right)}_{i=1,2,\ldots ,N}$. We are interested in measuring the heterogeneity of *X* with respect to a continuous latent representation *Z* defined on event space $Z=R^{2}$. In the present example, this space is simply the continuous 2-dimensional compression of an image that best facilitates its reconstruction. We choose a dimensionality of $n_{z}=2$ for the latent space in order to facilitate a pedagogically useful visualization of the latent feature representation, below. Unlike Section 4.1, in the present case we have no explicit representation of the true marginal distribution over the data, p(x).

Having defined the observable and latent spaces, measuring RRH now requires defining a model f:X→P(Z) that maps a (flattened) image vector $x_{i}\in X$ onto a probability distribution over the latent space. Our chosen model is the encoder module of a pre-trained convolutional variational autoencoder (cVAE) provided by the (https://colab.research.google.com/github/smartgeometry-ucl/dl4g/blob/master/variational_autoencoder.ipynb, Smart Geometry Processing Group at University College London) (Figure 7) [23,24]:
(55)$$f_{\phi }\left(z|x_{i}\right)=N\left(z|m\left(x_{i}\right),C\left(x_{i}\right)\right)$$
where ϕ are the encoder’s parameters, which specify a convolutional neural network (CNN) whose output layer returns a 2×1 mean vector $m(x_{i})$ and a 2×1 log-variance vector $s(x_{i})$ given $x_{i}$. For simplicity, we denote the latter as the 2×2 diagonal covariance matrix $C\left(x_{i}\right)={\left(e^{s_{j}\left(x_{i}\right)}\delta_{jk}\right)}_{j=1,2}^{k=1,2}$. Further details of the cVAE and its training can be found in Kingma and Welling [23,24], although the specific implementation in this paper was a pre-trained implementation by the (https://colab.research.google.com/github/smartgeometry-ucl/dl4g/blob/master/variational_autoencoder.ipynb, Smart Geometry Processing Group at University College London). Briefly, the cVAE learns to generate a compressed latent representation (via encoder $f_{\phi }$, which is an approximate posterior distribution) that contains enough information about the input $x_{i}$ to facilitate its reconstruction by a “decoder” module. The objective function is a lower bound on the model evidence p(x), which if maximized is equivalent to minimizing the Kullback–Leibler divergence between the approximate and true (but unknown) posteriors $f_{\phi }$ and p(z|x), respectively.

The continuous RRH under the model in Equation (Equation 55) for a single example $x_{i}\in X$ can be computed by merely evaluating the Rényi heterogeneity of a multivariate Gaussian (Equation (Equation 43) in Example 2) for the covariance matrix given by $C(x_{i})$. This is interpreted as the effective area of the 2-dimensional latent space consumed by representation of $x_{i}$.

Since the handwritten digit images belong to groups of “Zeros, Ones, Twos, …, Nines,” this section will call the quantity $\Pi_{q}^{W}$ the within-observation heterogeneity (rather than the “within-group” heterogeneity) in order to avoid its interpretation as measuring the heterogeneity of a group of digits. Rather, it is interpreted as the effective area of latent space consumed by representation of a single observation x∈X on average. It is computed by evaluation of Equation (Equation 44) at $C\left(X\right)={\left\{C(x_{i})\right\}}_{i=1,2,\ldots ,N}$, given uniform weights on samples.

Finally, to compute the pooled heterogeneity $\Pi_{q}^{P}$, we use the parametric pooling approach detailed in Example 2, wherein the pooled distribution is a multivariate Gaussian with mean and covariance given by Equations (Equation 41) and (Equation 42), respectively. The pooled heterogeneity is then merely Equation (Equation 43) evaluated at $C_{*}\left(X\right)$, and represents the total amount of area in the latent space consumed by the representation of *X* under $f_{\phi }$. The effective number of observations in *X* with respect to the continuous latent representation *Z* is, therefore, given by the between-observation heterogeneity:(56)$$\Pi_{q}^{B}\left(C(X),w\right)=\frac{\Pi_{q}^{P}\left(C_{*}\left(X\right)\right)}{\Pi_{q}^{W}\left(C(X),w\right)}.$$

Equation (Equation 56) gives the effective number of observations in *X* because it uses the entire sample X (of course, assuming X provides adequate coverage of the observable event space). However, one could compute the effective number of observations in a subset of X, if necessary. Let $X^{\left(j\right)}={\left(x_{k}\right)}_{k=1,2,\ldots ,N_{j}}$ be the subset of $N_{j}$ points in X found in the observable subspace $X_{j}\subset X$ (such as the subspace of MNIST digits corresponding to a given digit class). Given corresponding weights $w^{\left(j\right)}={\left(N_{j}^{-1}\right)}_{k=1,2,\ldots ,N_{j}}$, Equation (Equation 56) is then simply
(57)$$\Pi_{q}^{B}\left(C\left(X^{\left(j\right)}\right),w^{\left(j\right)}\right)=\frac{\Pi_{q}^{P}\left(C_{*}\left(X^{\left(j\right)}\right)\right)}{\Pi_{q}^{W}\left(C\left(X\right),w^{\left(j\right)}\right)}.$$

Figure 8 shows the effective number of observations in the subsets of MNIST images belonging to each image class, under the continuous representation learned by the cVAE. One can appreciate that the MNIST class of “Ones” (in the training set) has the smallest effective number of observations. Subjective visual inspection of the MNIST samples in Figure 6 may suggest that the Ones are indeed relatively more homogeneous as a group than the other digits (this claim is given further objective support in Appendix D based on deep similarity metric learning [47,48]).

Figure 9 demonstrates the correspondence of between-observation heterogeneity (i.e., the effective number of observations) and the visual diversity of different samples from the latent space of our cVAE model. For each image in the MNIST training dataset, we computed the effective location of its latent representation: $m(x_{i})$ for i∈{1,2,…,N}. For each of these image representations, we defined a “neighborhood” including the 49 other images whose latent coordinates were closest in Euclidean distance (which is sensible on the latent space given the Gaussian prior). For all such neighbourhoods defined, we then reconstructed the corresponding images on X, whose between-observation heterogeneity was then computed using Equation (Equation 57). Figure 9b shows the estimated effective number of observations for the latent neighborhoods with the greatest and least heterogeneity. One can appreciate that neighborhoods with $\Pi_{q}^{B}$ close to 1 include images with considerably less diversity than neighborhoods with $\Pi_{q}^{B}$ closer to the upper limit of 49. These data suggest that the between-observation heterogeneity—which is the effective number of observations in *X* with respect to the latent features learned by a cVAE—can indeed correspond to visually appreciable sample diversity.

## 5. Discussion

This paper introduced representational Rényi heterogeneity, a measurement approach that satisfies the replication principle [1,38,41] and is decomposable [28] while requiring neither a priori (A) categorical partitioning nor (B) specification of a distance function on the input space. Rather, the experimenter is free to define a model that maps observable data onto a semantically relevant domain upon which Rényi heterogeneity may be tractably computed, and where a distance function need not be explicitly manipulated. These properties facilitate heterogeneity measurements for several new applications. Compared to state-of-the-art comparator indices under a beta mixture distribution, RRH more reliably quantified the number of unique mixture components (Section 4.1), and under a deep generative model of image data, RRH was able to measure the effective number of distinct images with respect to latent continuous representations (Section 4.2). In this section, we further synthesize our conclusions, discuss their implications, and highlight open questions for future research.

The main problem we set out to address was that all state of the art numbers equivalent heterogeneity measures (Section 2.2) require a priori specification of a distance function and categorical partitioning on the observable space. To this end, we showed that RRH does not require categorical partitioning of the input space (Section 3). Although our analysis under the two-component BMM assumed that the number of components was known, RRH was the only index able to accurately identify an effectively singular cluster (i.e., where mixture components overlapped; Figure 5). We also showed that the categorical RRH did not violate the principle of transfers [39,40] (i.e., it was strictly concave with respect to mixture component weights), unlike the functional Hill numbers (Figure 5). Future studies should extend this evaluation to mixtures of other distributional forms in order to better characterize the generalizability of our conclusions.

Section 3.1 and Section 3.2 both showed that RRH does not require specification of a distance function on the observable space. Instead, one must specify a model that maps the observable space onto a probability distribution over the latent representation. This is beneficial since input space distances are often irrelevant or misleading. For example, latent representations of image data learned by a convolutional neural network will be robust to translations of the inputs since convolution is translation invariant. However, pairwise distances on the observable space will be exquisitely sensitive to semantically irrelevant translations of input data. Furthermore, semantically relevant information must often be learned from raw data using hierarchical abstraction. Ultimately, when (A) pre-defined distance metrics are sensitive to noisy perturbations of the input space, or (B) the relevant semantic content of some input data is best captured by a latent abstraction, the RRH measure will be particularly useful.

The requirement of specifying a representational model f:X→P(Z) implies the additional problem of model selection. In Section 3, we noted that the determination of whether a model is appropriate must be made in a domain-specific fashion. For instance, the method by which ecologists assign species labels prior to measurement of species diversity implies the use of a mapping from the observable space of organisms to a degenerate distribution over species labels (Example 1). In Section 4.2, we used the encoder module of a cVAE (a generative model based on a convolutional neural network architecture [23,24]) to represent images as 2-dimensional real-valued vectors in order to demonstrate our ability to capture variation in digits’ written forms (see Figure 7B and Figure 9). Someone concerned with measuring heterogeneity of image batches in terms of the digit-class distribution could choose a categorical latent representation corresponding to the digit classes (this would return the effective number of digit classes per sample). Regardless, the model used to map between observations and the latent space should be validated using either explanatory power (e.g., maximization of a lower bound on the model evidence), generalizability (e.g., out of sample predictive power), or another approach that is justifiable within the investigator’s scientific domain of interest.

In addition to the results of empirical applications of RRH in Section 4, we were also able to show that RRH generalizes the process by which species diversity and indices of economic equality are computed (Example 1). In doing so, we are able to clarify some of the assumptions inherent in those indices. Specifically, that assignment of species or ownership labels (in ecological and economic settings, respectively) corresponds to mapping from an observable space, such as the space of organisms’ identifiable features or the space of economic resources, onto a degenerate distribution over the categorical labels (Table 2). It is possible that altering the form of that mapping may yield new insights about ecological and economic diversity.

In conclusion, we have introduced an approach for measuring heterogeneity that requires neither (A) categorical partitioning nor (B) distance measure on the observable space. Our RRH method enables measurement of heterogeneity in disciplines where categorical entities are unreliably defined, or where relevant semantic content of some data is best captured by a hierarchical abstraction. Furthermore, our approach includes many existing heterogeneity indices as special cases, while facilitating clarification of many of their assumptions. Future work should evaluate the RRH in practice and under a broader array of models.

## Figures and Tables

**Figure 1 entropy-22-00417-f001:**
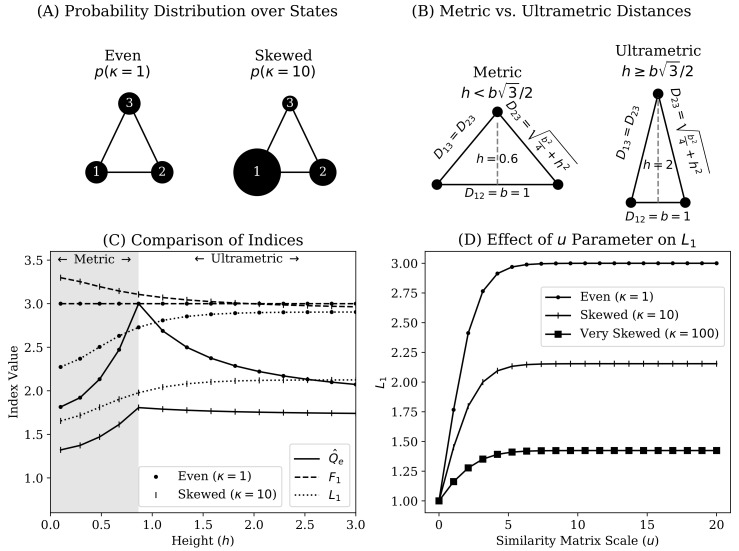
Illustration of simple three-state system under which we compare existing non-categorical heterogeneity indices. **Panel A** depicts a three state system *X* as an undirected graph, with node sizes corresponding to state probabilities governed by Equation (Equation 24). As 0≤κ diverges further from κ=1, the probability distribution over states becomes more unequal. **Panel B** visually represents the parametric pairwise distance matrix D(h,b) of Equation (Equation 25) (*h* is height, *b* is base length, $D_{ij}$ is distance between states *i* and *j*). In the examples shown in Panels B and C, we set b=1. Specifically, we provide visual illustration of settings for which the distance function on *X* is a metric (Definition 1; when $h<b\sqrt{3}/2$) or ultrametric (Definition 2; when $h\geq b\sqrt{3}/2$). **Panel C** compares the numbers equivalent quadratic entropy (solid lines marked ${\hat{Q}}_{e}$; Section 2.2.1), functional Hill numbers (at q=1, dashed lines marked $F_{1}$; Section 2.2.2), and the Leinster–Cobbold Index (at q=1, dotted lines marked $L_{1}$; Section 2.2.3) for reporting the heterogeneity of *X*. The y-axis reports the value of respective indices. The x-axis plots the height parameter for the distance matrix D(h,1) (Equation (Equation 25) and Panel B). The range of *h* at which D(h,1) is only a metric is depicted by the gray shaded background. The range of *h* shown with a white background is that for which D(h,1) is ultrametric. For each index, we plot values for a probability distribution over states that is perfectly even (κ=1; dotted markers) or skewed (κ=10; vertical line markers). **Panel D** shows the sensitivity of the Leinster–Cobbold index ($L_{1}$; y-axis) to the scaling parameter 0≤u (x-axis) used to transform a distance matrix into a similarity matrix ($S_{ij}=e^{-uD_{ij}}$). This is shown for three levels of skewness for the probability distribution over states (no skewness at κ=1, dotted markers; significant skewness at κ=10, vertical line markers; extreme skewness at κ=100, square markers).

**Figure 2 entropy-22-00417-f002:**
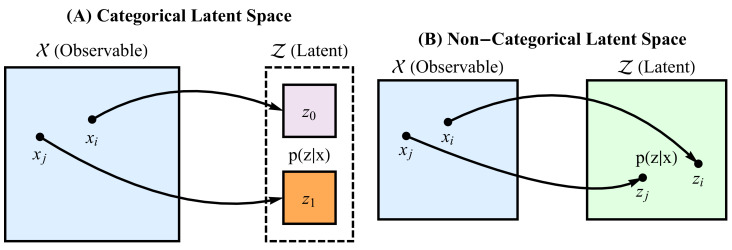
Graphical illustration of the two main approaches for computing representational Rényi heterogeneity. In both cases, we map sampled points on an observable space X onto a latent space Z, upon which we apply the Rényi heterogeneity measure. The mapping is illustrated by the curved arrows, and should yield a posterior distribution over the latent space. **Panel A** shows the case in which the latent space is categorical (for example, discrete components of a mixture distribution on a continuous space). **Panel B** illustrates the case in which the latent space has non-categorical topology. A special case of the latter mapping may include probabilistic principal components analysis. When the latent space is continuous, we must derive a parametric form for the Rényi heterogeneity.

**Figure 3 entropy-22-00417-f003:**
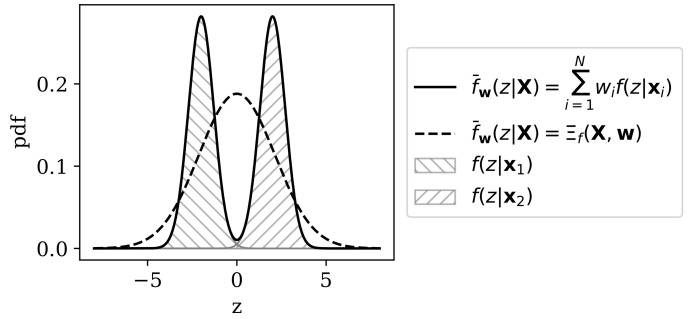
Illustration of approaches to computing the pooled distribution on a simple representational space Z=R. In this example, two points on the observable space, $(x_{1},x_{2})\in X$, are mapped onto the latent space via model $f(\cdot |x_{i})$ for i∈{1,2}, which indexes univariate Gaussians over Z (depicted as hatched patterns for $x_{1}$ and $x_{2}$, respectively). A pooled distribution computed non-parametrically by model-averaging (Equation (Equation 35)) is depicted as the solid black line. The parametrically pooled distribution (see Example 2) is depicted as the dashed black line. The parametric approach implies the assumption that further samples from X would yield latent space projections in some regions assigned low probability by $f(z|x_{1})$ and $f(z|x_{2})$.

**Figure 4 entropy-22-00417-f004:**
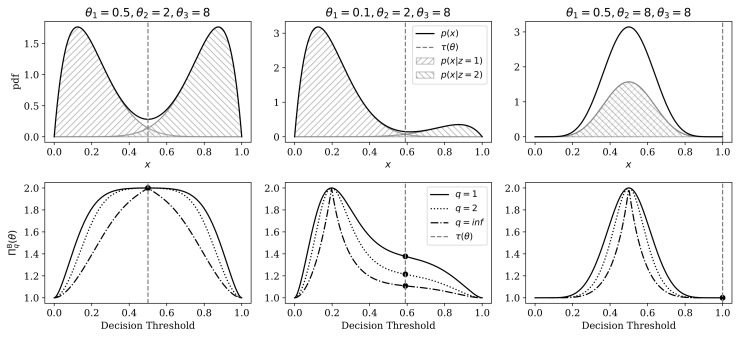
Demonstration of data-generating distribution (top row; Equations (Equation 45)–(Equation 47)), and relationship between the representational model’s decision threshold (Equations (Equation 48) and (Equation 50)) and categorical representational Rényi heterogeneity (bottom row). The optimal decision boundary (Equation (Equation 50)) is shown as a gray vertical dashed line in all plots. Each column depicts a specific parameterization of the data-generating system (parameters are stated above the top row). **Top Row:** Probability density functions for data-generating distributions. Shaded regions correspond to the two mixture components. Solid black lines denote the marginal distribution (Equation (Equation 47)). The x-axis represents the observable domain, which is the (0,1) interval. **Bottom Row:** Effect of varying categorical representational Rényi heterogeneity (RRH) for q∈{1,2,∞} across different category assignment thresholds for the beta-mixture models shown in the top row. Varying levels of decision boundary are plotted on the x-axis. The y-axis shows the resulting between-observation RRH. Black dots highlight the RRH computed at the optimal decision boundary.

**Figure 5 entropy-22-00417-f005:**
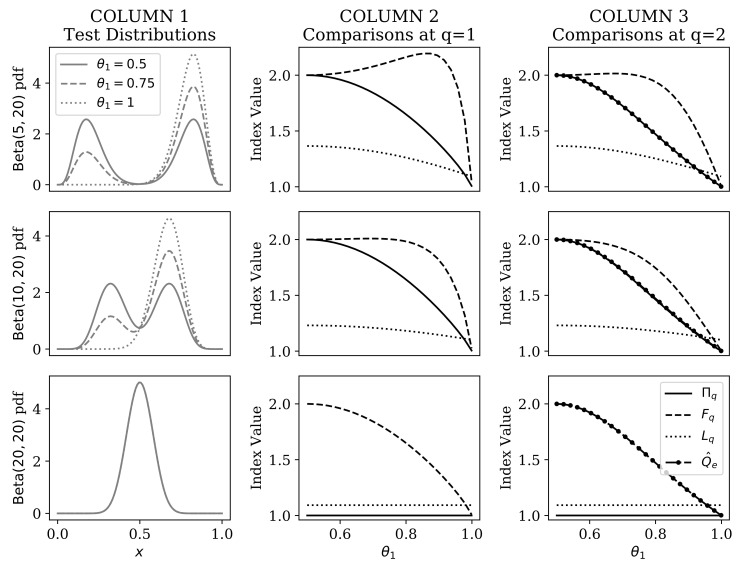
Comparison of categorical representational Rényi heterogeneity ($\Pi_{q}$), the functional Hill numbers ($F_{q}$), the numbers equivalent quadratic entropy (${\hat{Q}}_{e}$), and the Leinster–Cobbold index ($L_{q}$) within the beta mixture model. Each row of plots corresponds to a given separation between the beta mixture components. **Column 1** illustrates the beta mixture distributions upon which indices were compared. The x-axis plots the domain of the distribution (open interval between 0 and 1). The y-axis shows the corresponding probability density. Different line styles in Column 1 provides visual examples of the effect of changing the $\theta_{1}$ parameter over the range [0.5,1]. **Column 2** compares $\Pi_{q}$ (solid line), $F_{q}$ (dashed line), and $L_{q}$ (dotted line), each at elasticity q=1. The x-axis shows the value of the $0.5\leq \theta_{1}<1$ parameter at which the indices were compared. Index values are plotted along the y-axis. **Column 3** compares the indices shown in Column 2, as well as ${\hat{Q}}_{e}$ (dot-dashed line).

**Figure 6 entropy-22-00417-f006:**
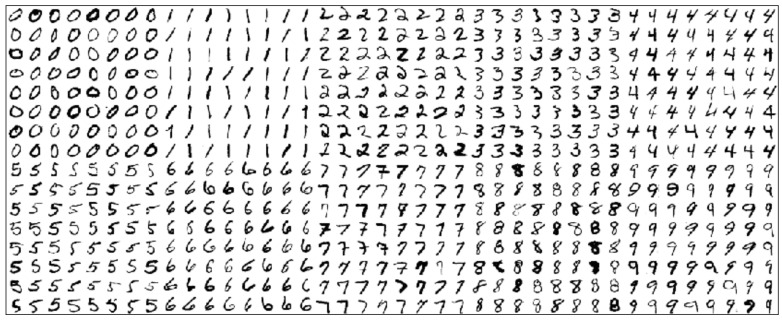
Sample images from the MNIST dataset [22].

**Figure 7 entropy-22-00417-f007:**
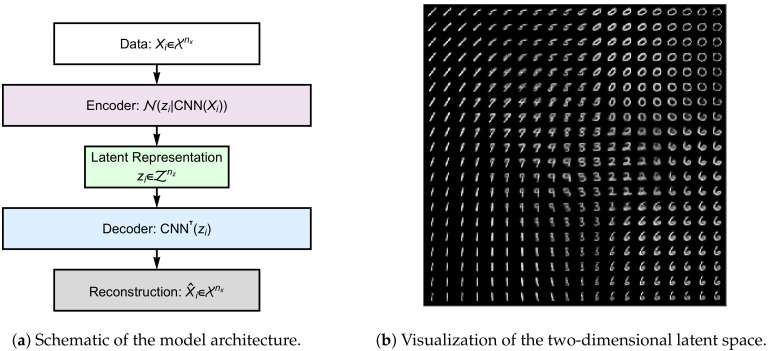
**Panel A**: Illustration of the convolutional variational autoencoder (cVAE) [Equation 23]. The computational graph is depicted from top to bottom. An *n_x_*-dimensional input data ***X**_i_* (white rectangle) is passed through an encoder (in our experiment this is a convolutional neural network, CNN) which parameterizes an *n_z_*-dimensional multivariate Gaussian over the coordinates ***z**_i_* for the image’s embedding on the latent space $Z=R^{2}$. The latent embedding can then be passed through a decoder (blue rectangle) which is a neural network employing transposed convolutions (here denoted CNN^⊤^) to yield a reconstruction ${\hat{X}}_{i}$ of the original input data. The objective function for this network is a variational lower bound on the model evidence of the input data (see Kingma and Welling [Equation 23] for details). **Panel B:** Depiction of the latent space learned by the cVAE. This model was a pre-trained model from the (https://colab.research.google.com/github/smartgeometry-ucl/dl4g/blob/master/variational_autoencoder.ipynb, Smart Geometry Processing Group at University College London).

**Figure 8 entropy-22-00417-f008:**
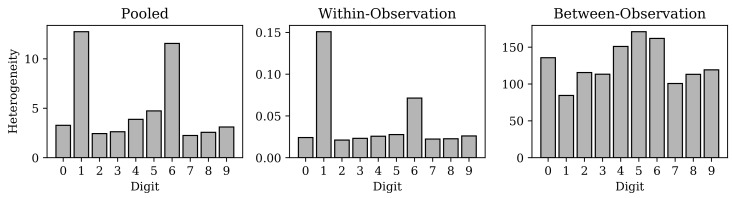
Heterogeneity for the subset of MNIST training data belonging to each digit class respectively projected onto the latent space of the convolutional variational autoencoder (cVAE). The leftmost plot shows the pooled heterogeneity for each digit class (the effective total area of latent space occupied by encoding each digit class). The middle plot shows the within-observation heterogeneity (the effective total area of latent space per encoded observation of each digit class, respectively). The rightmost plot shows the between-observation heterogeneity (the effective number of observations per digit class). Recall that Rényi heterogeneity on a continuous distribution gives the effective size of the domain of an equally heterogeneous uniform distribution on the same space, which explains why the within-observation heterogeneity values here are less than 1.

**Figure 9 entropy-22-00417-f009:**
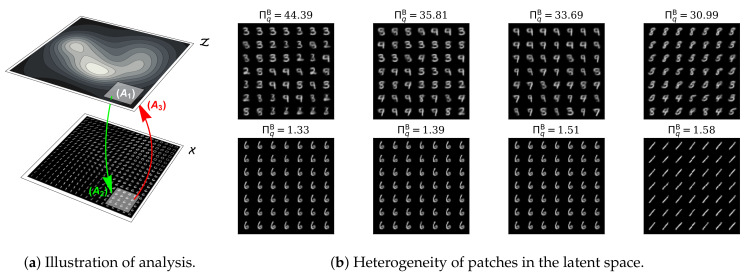
Visual illustration of MNIST image samples corresponding to different levels of representational Rényi heterogeneity under the convolutional variational autoencoder (cVAE). **Panel (a)** illustrates the approach to this analysis. Here, the surface Z shows hypothetical contours of a probability distribution over the 2-dimensional latent feature space. The surface X represents the observable space, upon which we have projected an “image” of the latent space Z for illustrative purposes. We ﬁrst compute the expected latent locations **m**(***x**_i_*) for each image
$x_{i}\in X$
**(A_1_)** We then deﬁne the latent neighbourhood of image ***x**_i_* as the 49 images whose latent locations are closest to **m**(***x**_i_*) in Euclidean distance. **(A_2_)** Each coordinate in the neighbourhood of **m**(***x**_i_*) is then projected onto a corresponding patch on the observable space of images. **(A_3_)** These images are then projected as a group back onto the latent space, where Equation (57) can be applied, given equal weights over images, to compute the effective number of observations in the neighbourhood of ***x**_i_*. **Panel (b)** plots the most and least heterogeneous neighbourhoods so that we may compare the estimated effective number of observations with the visually appreciable sample diversity.

**Table 1 entropy-22-00417-t001:** Relationships between Rényi heterogeneity and various diversity or inequality indices for a system *X* with event space X={1,2,…,n} and probability distribution $p={\left(p_{i}\right)}_{i=1,2,\ldots ,n}$. The function 𝟙[·] is an indicator function that evaluates to 1 if its argument is true or to 0 otherwise.

Index	Expression
Observed richness [31]	$\Pi_{0}\left(p\right)=\sum_{i=1}^{n}𝟙\left[p_{i}>0\right]$
Perplexity [30]	$\Pi_{1}\left(p\right)=\exp\left\{-\sum_{i=1}^{n}p_{i}\log p_{i}\right\}$
Inverse Simpson concentration [1]	$\Pi_{2}\left(p\right)={\left(\sum_{i=1}^{n}p_{i}^{2}\right)}^{-1}$
Berger-Parker Diversity Index [32,33]	$\Pi_{\infty }\left(p\right)={\left(\max_{i}p_{i}\right)}^{-1}$
Rényi entropy [18]	$R_{q}\left(p\right)=\log\Pi_{q}\left(p\right)$
Shannon entropy [29]	$H\left(p\right)=\log\Pi_{1}\left(p\right)$
Tsallis entropy [34]	$T_{q}\left(p\right)=\frac{1}{q-1}\left(1-\Pi_{q}{\left(p\right)}^{1-q}\right)$
Simpson concentration [35]	$\mathrm{Simpson}\left(p\right)={\left(\Pi_{2}\left(p\right)\right)}^{-1}$
Gini-Simpson index [36]	GSI(p)=1−Simpson(p)
Generalized entropy index [3,37]	$\mathrm{GEI}\left(p\right)=\frac{1}{q(q-1)}\left[{\left(\frac{1}{n}\Pi_{q}\left(p\right)\right)}^{1-q}-1\right]$

**Table 2 entropy-22-00417-t002:** Definitions in formulation of classical biodiversity and economic equality analysis as categorical representational Rényi heterogeneity. Superscripted indexing on $x={\left(x_{i}\right)}^{i=1,\ldots ,n_{x}}$ denotes that this is a row vector.

	Analytical Context	
Symbol	Biodiversity	Economic Equality
*X*	Ecosystem, whose observation yields an organism denoted by vector $x={\left(x_{i}\right)}^{i=1,\ldots ,n_{x}}\in X$	A system of resources, whose observation yields an asset denoted by vector $x={\left(x_{i}\right)}^{i=1,\ldots ,n_{x}}\in X$
$X\subseteq R^{n_{x}}$	$n_{x}$-dimensional feature space of organisms in the ecosystem	$n_{x}$-dimensional feature space of assets in the economy, whose topology is such that the “economic” or monetary value is equal at each coordinate x∈X
$Z=\left\{z\in {\left\{0,1\right\}}^{n_{z}}:\sum_{i=1}^{n_{z}}z_{i}=1\right\}$	$n_{z}$-dimensional space of one-hot species labels	$n_{z}$-dimensional space of one-hot labels over wealth-owning agents
f:X→P(Z)	A model that performs the mapping x↦f(x) of organisms to discrete probability distributions over Z	A model that performs the mapping x↦f(x) of assets to discrete probability distributions over Z
$N_{i}\in N_{+}$	The number of organisms observed belonging to species $i\in \left\{1,\ldots ,n_{z}\right\}$	The number of equal valued assets belonging to agent $i\in \left\{1,\ldots ,n_{z}\right\}$
$N=\sum_{i=1}^{n_{z}}N_{i}$	The total number of organisms observed	The total quantity of assets observed
$X={\left(x_{ij}\right)}_{i=1,\ldots ,N}^{j=1,\ldots ,n_{x}}$	A sample of *N* organisms	A sample of *N* assets
$w={\left(w_{i}\right)}_{i=1,\ldots ,N}$	Sample weights, such that $w_{i}\geq 0$ and $\sum_{i=1}^{N}w_{i}=1$	

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
