# Peer review of "Representational Rényi Heterogeneity"

_entropy, 2020, doi:10.3390/e22040417_

Round 1
Reviewer 1 Report
I really enjoyed reading this revised version of the paper.
Maths are now clear and the paper seems ready to go.
A few minor comments:
- Equation (28), $x_i$ should be bold.
- A bit confused in line 371--372. I thought we first define $f$, and then compute $\Pi_q$ (maybe it does not matter)
- line 601, you may want to include zeros
The one extra comment (or just question) is regarding the choice of function $f$. The heterogeneity depends on $f$ which may vary with network structure (in MNIST example) or training set, etc. Moreover, we can use categorical representation in the same MNIST example with a regular CNN network with a softmax layer. The author may want to mention the way of finding an appropriate choice of function $f$ (or at least intuition).
Author Response
Thank you for your helpful suggestions. Please see the attached file for associated revisions.

Reviewer 2 Report
The paper is far more clear and almost ready for publication. The improvement is spectacular since the first version.
I only have a few minor concerns left.
Section 2.1 and along the paper: the introduction of Rényi heterogenity you chose is quite intuitive but has two drawbacks: it only allows q to be a positive integer and it does not work for q=1 (see eq. 3 for instance). Actually, q can be any real number. You should write it in section 2.1: explain you chose this approach because it is very intuitive and you only use integer q values in the paper but can be generalized to any value of q, cite Jost 2007. The problem of q=1 is more serious because Shannon's entropy can't be disregarded. You have to explain that all your results are valid at q=1 by continuity (cite Jost again for eq. 3) after you have justified that q can be real and that you won't write the special formulas at q=1 for conciseness.
This may not be so easy for the proofs in appendices.
Eq. 7: Jost defined an ad-hoc partitioning to ensure independence between within and between diversity. It is highly arguable since it does not follow the usual definitions of between diversity, see e.g. Tuomisto (2010) and, more problematically, within-diversity as defined by Pattil and Taillie (1982). Weights should not be set to power q in eq. 7 or within diversity is not the (arithmetic) average of group diversities. This is a serious issue since, for example, the equivalence between variance partitioning and diversity partitioning falls.
This idea has not been followed much, at least in ecology. A better definition is simply with weights at power 1. It allows a correct partitioning of diversity and the respect of the classical definition of within diversity (Marcon et al., 2014). The good news is that, since you use equally-weighted groups along the paper, results are identical.
Line 552: It is not obvious to understanf why the original X space is of dimension 784. Please explain that images are 28x28 black or white points for readers who are not familiar with the dataset.
References
Marcon, E., Scotti, I., Hérault, B., Rossi, V., & Lang, G. (2014). Generalization of the Partitioning of Shannon Diversity. Plos One, 9(3), e90289. https://doi.org/10.1371/journal.pone.0090289
Patil, G. P., & Taillie, C. (1982). Diversity as a concept and its measurement. Journal of the American Statistical Association, 77(379), 548–561. https://doi.org/10.2307/2287709
Tuomisto, H. (2010). A diversity of beta diversities: straightening up a concept gone awry. Part 1. Defining beta diversity as a function of alpha and gamma diversity. Ecography, 33(1), 2–22. https://doi.org/10.1111/j.1600-0587.2009.05880.x
Author Response
Thank you for your helpful suggestions. Please see the attached file for respective revisions.

This manuscript is a resubmission of an earlier submission. The following is a list of the peer review reports and author responses from that submission.
Round 1
Reviewer 1 Report
The paper introduces Representational Rényi Heterogeneity to deal with ill-defined categories or lacking a distance measure between categories.
The main argument of the paper is that original data can be mapped to a latent space where heterogeneity can be calculated with classical metrics, I.E. Hill numbers.
The method is applied to two example cases, a beta mixture model where original data is mapped into a binary space and a dataset of handwritten digit images mapped into the space produced by a variational autoencoder.
Representational Rényi Heterogeneity is a really interesting approach and the paper contains a lot of interesting things but I was not convinced by it as a whole for several reasons.
First, it is built around to examples that are hard to generalize. The wide-interest part of the paper is fig. 4, which explains the concept. The problem is that there is no universal mapping to be used so the reader is not much more equipped to deal with his own datasets after reading the paper. I'm afraid this is a fundamental limit without any solution. Then, the paper should be much more concise and avoid the very detailed analysis of the handwritten-digit dataset, worth another paper. In summary, the purpose of the paper and its plan do not fit : it is made to introduce a new method, does it well in a few lines (introduction and fig. 4) but uses most of its space to study a particular example with little general interest.
Second, it is not well organized. Section 2 reviews existing methods, but introduces a new one \hat{Q}_e, which is out of the scope of the paper: previous methods are used as a benchmark.
In section 2, the decomposition of Hill numbers is not introduced, but it appears is section 3.2. Starting from eq. 28, the between-observation heterogeneity is focused on, so it should have been explained in depth, in section 2.
Third, it is often quite cryptic.
Fig.1, panel A claims to be a demonstration of the cases where the matrix D(h,b) is or is not ultrametric. This is not obvious! In the text, the form of the matrix should be explained. h is for height, but you have to read the legend of fig.1 to understand it. I guess b is for base, but this is not written anywhere.
As I understand it, an important parameter of the CVAE is the dimension of the latent space. It is set to 2 (line 222), without much explanation. Why not 10, since there are 10 digits? A clearer, more intuitive presentation of the latent space is necessary for the reader to understand it precisely. I was not able to really understand section 3.2.3 because of too much uncertainty about what the latent space contains.
Last, I am really troubled by the values of within-observation heterogeneity in fig.8. You cite [26] and [27] to introduce number equivalent. These authors only address discrete probability distributions. At least in this case, number equivalents are at least 1 (perfect homogeneity). I have little experience of continuous distributions, but I do not see how within-observation heterogeneity could be <1. This must be checked, and, if correct, explained.
Reviewer 2 Report
Please check the review report (attached).
